# Glomerular Mesangial Cell pH Homeostasis Mediates Mineralocorticoid Receptor-Induced Cell Proliferation

**DOI:** 10.3390/biomedicines9091117

**Published:** 2021-08-30

**Authors:** Michael Gekle, Sigrid Mildenberger

**Affiliations:** Julius-Bernstein-Institut für Physiologie, Martin-Luther-Universität Halle-Wittenberg, 06112 Halle (Saale), Germany; sigrid.mildenberger@medizin.uni-halle.de

**Keywords:** aldosterone, mesangial cell, pH, proliferation

## Abstract

Mineralocorticoids (e.g., aldosterone) support chronic inflammatory tissue damage, including glomerular mesangial injury leading to glomerulosclerosis. Furthermore, aldosterone leads to activation of the extracellular signal-regulated kinases (ERK1/2) in rat glomerular mesangial cells (GMC). Because ERK1/2 can affect cellular pH homeostasis via activation of Na^+^/H^+^-exchange (NHE) and the resulting cellular alkalinization may support proliferation, we tested the hypothesis that aldosterone affects pH homeostasis and thereby cell proliferation as well as collagen secretion also in primary rat GMC. Cytoplasmic pH and calcium were assessed by single-cell fluorescence ratio imaging, using the dyes BCECF or FURA2, respectively. Proliferation was determined by cell counting, thymidine incorporation and collagen secretion by collagenase-sensitive proline incorporation and ERK1/2-phosphorylation by Western blot. Nanomolar aldosterone induces a rapid cytosolic alkalinization which is prevented by NHE inhibition (10 µmol/L EIPA) and by blockade of the mineralocorticoid receptor (100 nmol/L spironolactone). pH changes were not affected by inhibition of HCO_3_^−^ transporters and were not dependent on HCO_3_^−^. Aldosterone enhanced ERK1/2 phosphorylation and inhibition of ERK1/2-phosphorylation (10 µmol/L U0126) prevented aldosterone-induced alkalinization. Furthermore, aldosterone induced proliferation of GMC and collagen secretion, both of which were prevented by U0126 and EIPA. Cytosolic calcium was not involved in this aldosterone action. In conclusion, our data show that aldosterone can induce GMC proliferation via a MR and ERK1/2-mediated activation of NHE with subsequent cytosolic alkalinization. GMC proliferation leads to glomerular hypercellularity and dysfunction. This effect presents a possible mechanism contributing to mineralocorticoid receptor-induced pathogenesis of glomerular mesangial injury during chronic kidney disease.

## 1. Introduction

Activation of the mineralocorticoid receptor promotes chronic inflammatory injury of non-epithelial tissue resulting in cardiovascular and renal damage [1,2,3,4,5,6,7]. Administration of the mineralocorticoid aldosterone to rats leads to glomerular injury with mesangial matrix expansion, cell overgrowth and proinflammatory signaling [8,9]. Furthermore, aldosterone stimulates glomerular mesangial cell (GMC) proliferation in culture [10]. Thus, GMC proliferation may contribute to aldosterone-induced renal injury [11].

The classical mechanism of aldosterone action involves binding to its intracellular receptor, transcription and protein synthesis. However, aldosterone can also induce rapid responses including intracellular pH and calcium, intracellular generation of inositol-1,4,5-trisphosphate (IP_3_) and protein kinase C activation [12,13,14,15,16,17,18,19,20,21,22]. A central common theme of rapid aldosterone action is furthermore activation of the Na^+^/H^+^-exchanger via extracellular signal-regulated kinases (ERK1/2) [23]. Several studies provide evidence that at least part of these rapid effects is mediated by the mineralocorticoid receptor (MR) [10,23,24,25,26,27,28,29,30,31]. Nishiyama et al. [10] showed rat glomerular mesangial cells (GMC) respond to aldosterone with a rapid activation of ERK1/2, which is required for aldosterone-induced proliferation of GMC. This study also provided evidence that the rapid effect is mediated by the MR.

Cell proliferation depends on a proper cytoplasmic pH homeostasis, maintained to a large extent by the Na^+^/H^+^-exchanger type 1 (NHE1) [32,33]. This is also true for GMC [34]. NHE1 is activated by growth factors, to a large extent via ERK1/2 [33,35], resulting in cytoplasmic alkalinization, which supports proliferation. Aldosterone can also stimulate NHE1 in an ERK1/2-dependent manner, leading to cytoplasmic alkalinization [36,37]. In the present study, we tested the hypothesis that aldosterone-induced proliferation of GMC depends on ERK1/2- and NHE1-mediated alkalinization. 

## 2. Materials and Methods

### 2.1. Cell Culture

Rat glomerular mesangial cells (GMC) were prepared from male Sprague-Dawley rats by a sieving technique under sterile conditions. After collection of isolated glomeruli, “glomerular cores” were prepared by partial collagenase digestion and then plated in HEPES-buffered RPMI 1640 medium (Biochrom, Berlin, Germany) supplemented with 20% fetal calf serum (Merck KGaA, Darmstadt, Germany), 5 mg/L each of insulin (Sigma, St. Louis, MO, USA), transferrin (Sigma, St. Louis, MO, USA), as well as 5 µg/L sodium selenite (Sigma, St. Louis, MO, USA) at 37 °C in a humified 5% CO_2_-95% air mixture. After reaching confluence primary mesangial cells were subcultured every 7–10 days. GMC were identified by their typical morphology as well as by immunocytochemical and biochemical methods [38]. For the present study, GMC (passage 10–20) were maintained in RPMI 1640 medium supplemented with 10% fetal calf serum, 5 mg/L each of insulin, transferrin, as well as 5 µg/L sodium selenite at 37 °C in a humified 5% CO_2_-95% air mixture. After growth to confluence, cells were washed once and made quiescent by 48 h incubation in supplement-free medium.

### 2.2. Determination of Cytosolic pH

Intracellular pH of single cells was determined using the pH-sensitive dye BCECF (Molecular Probes, Thermo Fisher Scientific, Waltham, MA, USA) as described elsewhere [39,40,41]. In brief, cells were incubated with 2 × 10^−6^ mol/L BCECF-AM for 5 min. Then, the coverslips were rinsed 4 times with superfusion solution, transferred to the stage of an inverted Axiovert 100 TV microscope (Zeiss, Oberkochen, Germany) and allowed to equilibrate for 15 min. Excitation light source was a 100 W mercury lamp. The excitation wavelengths were 490 nm/450 nm. The emitted light was filtered through a bandpass-filter (515–565 nm). The data acquisition rate was one fluorescence intensity ratio every 2 s. Images were digitized online using a video-imaging software (Hamamatsu, Herrsching, Germany). pH calibration was performed after each experiment by the nigericin (Sigma, St. Louis, MO, USA) technique [39,40], using at least three calibration solutions in the range from pH 6.8 to 7.8. The calibration solutions contained 115 mmol/L KCl and 30 mmol/L NaCl. 

### 2.3. Determination of Cytosolic Free Calcium

Cytosolic free calcium was determined using the Ca^2+^ sensitive dye fura-2 (Molecular Probes, Thermo Fisher Scientific, Waltham, MA, USA) [42]. In brief, cells were incubated with fura-2 AM in a final concentration of 5 µmol/L for 15 min and mounted on the stage of an inverted Axiovert 100 TV microscope (400× magnification, oil immersion; Zeiss, Oberkochen, Germany). The fluorescence signal was monitored at 510 nm with excitation wavelength alternating between 334 and 380 nm using a 100 W xenon lamp. [Ca^2+^]_i_ was calculated according to [42] using a K_d_ of 225 nmol/L. The maximum and minimum ratios (R_max_ and R_min_) were measured after addition of calibration solutions, containing 1 µmol/L ionomycin (Sigma, St. Louis, MO, USA) and 1 mmol/L Ca^2+^ to determine R_max_ or 1 mmol/L EGTA and no Ca^2+^ for R_min_, respectively. Possible artifacts were excluded by measurement of autofluorescence without fura-2.

### 2.4. Determination of Cell Number

Cell number was determined using a Coulter Counter Z2 series (Beckman, Indianapolis, IN, USA). Cells were seeded on plastic dishes and grown to subconfluence (ca. 40,000 cells/cm^2^). Cells were made quiescent by removal of additives and serum for 24 h. The experiments were carried out in medium without additives, in order to ensure that growth stimulation was solely dependent on addition of aldosterone. 

### 2.5. Thymidine Incorporation

A 0.5 µL volume of the [^3^H]-thymidine stock solution (37 MBq/mL; Thermo Fisher Scientific, Waltham, MA, USA) was added to 1 mL of cell culture medium [43]. After 48 h the TCA-insoluble radioactivity was determined as follows: Cell were rinsed three times with phosphate-buffered saline at 4 °C, followed by one wash step with 10% TCA (15 min; Merck KGaA, Darmstadt, Germany) and two wash steps with 5% TCA (5 min) and a final wash step with ethanol. Subsequently, 0.5 mL NaOH (1 N) was added to each dish and incubated overnight. Thereafter, radioactivity was counted [43].

### 2.6. Determination of IP_3_ Formation

IP_3_ formation was determined by anion exchange columns [44]. In brief, Cells seeded in 6-well plates are preincubated during 24 h in media containing 0.5 µCi/mL [^3^H]-inositol (Thermo Fisher Scientific, Waltham, MA, USA). Prior to experimentation, radioactive medium was aspirated, the cells were washed with 3 × 2 mL HEPES-Ringer, and then incubated for 30 min in 2 mL HEPES-Ringer containing 15 mM LiCl (pH 7.4). This medium was replaced with 1 mL HEPES-Ringer + 15 mM LiCl and after 10 min of incubation at 37 °C 1 mL aliquots of control Ringer + 15 mM LiCl with desired agonists were added. After 15 min of incubation, cells were lysed with 1 mL of 4 mM EDTA/1% SDS (90 °C) and lysates were applied to ion exchange columns prepared as follows. An amount of 0.5 g of Dowex-AG 1X-8 (formate form; Bio-Rad, Munich, Germany) was laid into 5 mL pipette tips with cotton wool on the bottom. Applied samples were washed with 2 mL aliquots of H_2_O and 5 mM disodium tetraborate/60 mM sodium formate (Merck KGaA, Darmstadt, Germany) and then InsP_1_, InsP_2_ and InsP_3_ were eluted by subsequent addition of 2 mL of 0.1 M formic acid/0.2 M ammonium formate (InsP_1_), 0.1 M formic acid/0.4 M ammonium formate (InsP_2_) and 0.1 M formic acid/1.0 M ammonium formate (InsP_3_) (Merck KGaA, Darmstadt, Germany). Eluted InsP_1_-InsP_3_ were collected into scintillation vials, mixed with 10 mL of scintillation cocktail and counted.

### 2.7. Determination of Collagen Secretion by Collagenase-Sensitive Proline Incorporation

Total collagen secretion and total secreted protein were assessed by proline incorporation assay [45,46]. Cells were treated with aldosterone in the presence of 0.5 µCi/mL of [^3^H]proline (Thermo Fisher Scientific, Waltham, MA, USA), 50 mg/L ascorbic acid and 50 mg/L ß-aminoproprionitrile (Merck KGaA, Darmstadt, Germany). Determination of total secreted proteins or collagen was performed in the supernatants. Four hundred-microliter aliquots of supernatant from each well were incubated with 100 µL of collagenase assay buffer (50 mM Tris HCl, pH 7.5, 5 mM CaCl_2_, and 2.5 mM N-ethylmaleimide) containing 30 U/mL of collagenase (from Clostridium histolyticum, Sigma, St. Louis, MO, USA) for 4 h at 37 °C. In parallel, a second 400-µL aliquot was incubated in assay buffer without collagenase. Then, 50 µL of FBS and 100 µL of TCA (Merck KGaA, Darmstadt, Germany) were added to the samples and incubated on ice for 30 min to precipitate protein fractions. Precipitates were washed three times with 2 mL of TCA and two times with 2 mL of 80% ethanol. Finally, precipitates were dissolved in 1 M NaOH and radioactivity determined by liquid scintillation counting. Amounts of total secreted proteins were calculated as disintegrations per minute (dpm) in supernatants without collagenase. Secreted collagen was calculated as dpm in supernatants without collagenase minus dpm in supernatants with collagenase. The percentage of total protein secreted as collagen (%collagen) was calculated as the ratio of collagenase-releasable dpm divided by total dpm as follows: %collagen = (C/P)/[[5.4 × (1-C/P)] + (C/P)] × 100, where C is collagenase-releasable dpm in supernatants and P is collagenase-insensitive dpm in pellets. A correction factor of 5.4 for noncollagen protein was used to adjust for the relative abundance of proline and hydroxyproline in proteins containing collagen.

### 2.8. Western Blot

Western blotting was performed according to standard protocols. Cells were lysed (0.5 M Tris-HCl pH 6.8; 10% SDS; 10% 2-Mercaptoethanol; 20% Glycerol; 0.01% Bromphenolblue; Sigma, St. Louis, MO, USA), separated by SDS-PAGE and transferred to a nitrocellulose membrane. Subsequently, membranes were incubated with antibodies specific for phospho-ERK1/2 or total ERK1/2 (Cell Signaling, Frankfurt, Germany). The bound primary antibody was visualized using horseradish peroxidase (HRP)-conjugated secondary antibodies and Serva chemoluminescence reagent for HRP (Serva, Heidelberg, Germany) with the Molecular Imager ChemiDoc XRS System (Bio-Rad, Munich, Germany). Quantitative analysis was performed with Quantity One software, version 4.6.9 (Bio-Rad, Munich, Germany).

### 2.9. Materials

Aldosterone, 4,4′-diisothiocyanatostilbene-2,2′-disulfonic acid (DIDS) and ethyl-isopropanol-amiloride (EIPA) were obtained from Sigma (St. Louis, MO, USA), U0126 is from Tocris Cookson (Bristol, UK). The inhibitors were used at appropriate published concentrations EIPA [47,48], U0126 [49], DIDS [50] and BIM [51,52]. All other applied chemicals were of analytical grade and obtained from Merck, Darmstadt, Germany. HCO_3_^−^ Ringer solution was composed of (mmol/L): NaCl 108, KCl 5.4, CaCl_2_ 1.2, MgCl_2_ 0.8, NaH_2_PO_4_ 1.0, NaHCO_3_ 24, glucose 5.5, 5% CO_2_. HEPES-Ringer solution was composed of (mmol/L): NaCl 124.5, KCl 5.4, CaCl_2_ 1.2, MgCl_2_ 0.8, NaH_2_PO_4_ 1.0, HEPES 10, glucose 5.5, pH 7.4. Control solutions always contained the appropriate amount of vehicle (DMSO or ethanol, <1‰). 

### 2.10. Statistics

The data are presented as mean values ± SEM. Significance of difference was tested by paired or unpaired Student’s *t*-test or ANOVA, as applicable. Differences were considered significant if *p* < 0.05. Cells from at least three different passages were used for each experimental series. N represents the number of cells or tissue culture dish investigated. For pH and calcium determinations at least five cover slips from at least three different passages were investigated for each experimental condition.

## 3. Results

### 3.1. Aldosterone-Induced Changes in Cytosolic pH

As shown in Figure 1A, aldosterone exerts a rapid biphasic effect on cytosolic pH, similar to the one described in vascular smooth muscle cells [21]. In HEPES-buffered saline, addition of 10 nmol/L aldosterone induced an initial transient acidification followed by a sustained alkalinization (Figure 1A). A similar effect was observed in HCO_3_^−^/CO_2_-buffered saline (Figure 1B). In the presence of 1 nmol/L aldosterone, we observed a similar biphasic response (Figure 1C). 

To exclude that the observed effects represent spontaneous pH changes, we performed time controls. As shown in Figure 1D, the aldosterone-induced effects were not the result of spontaneous pH changes. We observed a slight decline in cytosolic pH over time, indicating that aldosterone-induced acidification might be slightly overestimated. Figure 1E summarizes the pH values under control conditions, during the acidification phase and during sustained alkalinization. Because cytosolic pH reaches a new steady state at more alkaline values, we hypothesized that H^+^-transport systems responsible for the control of cellular pH homeostasis are affected. The most likely candidate is the Na^+^/H^+^-exchanger, shown to be stimulated by aldosterone in other cell types [23].

### 3.2. Pharmacology of Aldosterone-Induced Changes in Cytosolic pH

To characterize the aldosterone-induced pH changes in more detail, we used a pharmacological approach. We tested the effect of EIPA (a well-established Na^+^/H^+^-exchange inhibitor [47]) and DIDS (an inhibitor of Cl^−^ and HCO_3_^−^ transport [53]). As shown in Figure 2A, EIPA (10 µmol/L) completely prevented aldosterone-induced alkalinization.

In contrast, DIDS (200 µmol/L, Figure 2B) was without effect. The results are summarized in Figure 2E. Aldosterone-induced acidification could not be prevented by either EIPA or DIDS. At present, the underlying mechanism for GMC acidification is not known.

Rapid activation of NHE1 can be mediated through ERK1/2 [33,47]. Because it had been shown that aldosterone activated ERK1/2 in GMC with a similar time course to that in which alkalinization occurs [10], which we could confirm as shown in Figure 3E, we tested whether inhibition of ERK1/2 activation interfered with the alkalinizing effect of aldosterone. Blockade of ERK1/2 activation with 1 µmol/L U0126 [54] prevented aldosterone-induced alkalinization (Figure 2C). Furthermore, alkalinization could be prevented by inhibition of protein kinase C with 200 nmol/L bisindolylmaleimide (BIM; Figure 2D). The results are summarized in Figure 2E. Aldosterone-induced acidification was not prevented by either U0126 or BIM. Previously, it has been reported that BIM exerts no effect on MR transcriptional activity [55]. By contrast, ERK1/2 is required for regular nuclear translocation [31], which leads to the modulation of transcriptional activity. By contrast, DIDS has been shown not to prevent transcriptional effects of aldosterone [56]. Similarly, EIPA did not prevent MR-induced upregulation of sodium channel expression [57].

### 3.3. Aldosterone-Induced Cell Proliferation and Collagen Secretion

Because ERK1/2 and alkalinization play a role in the regulation of cell proliferation, we investigated whether aldosterone affects proliferation of rat mesangial cells, and if so, whether this effect can be prevented by U0126 and EIPA. Figure 3 shows that 48 h incubation with aldosterone led to a significant increase in cell number and thymidine incorporation compared to control conditions. 

Because cell number was constant under control conditions (Figure 3A), these data show that aldosterone indeed induced proliferation and did not just prevent a reduction in cell number. Furthermore, the aldosterone-induced increase in cell number was prevented by U0126 (Figure 3B,C), indicating that ERK1/2 activation was necessary for proliferation, as already suggested before [10]. Proliferation was also inhibited by 10 µmol/L EIPA (Figure 3B,C). Thus, aldosterone-induced stimulation of NHE1 supports the proliferative action the hormone.

Aldosterone enhanced the amount of secreted collagen significantly (Figure 3D). If this effect were to occur in the intact glomerulum, it would promote glomerulosclerosis. In the presence of spironolactone, U0126 or EIPA, aldosterone did not enhance collagen secretion (Figure 3D). To determine whether enhanced extracellular collagen resulted from a specific stimulation of collagen secretion or was the result of a more general enhancement of protein secretion (due to the increased cell number), we determined the percentage of total protein secreted as collagen. Aldosterone did not affect this parameter significantly (21.02 ± 2.68% under control conditions versus 24.49 ± 2.64% in the presence of aldosterone; n = 8). These data imply that enhanced extracellular collagen was a result of the increase in cell number. Of course, these data do not rule out that less abundant types of collagen were induced specifically by aldosterone. 

### 3.4. No Contribution of Cytosolic Ca^2+^ to Aldosterone-Induced Alkalinisation or Cell Proliferation

Aldosterone exerted a minor effect on cytosolic Ca^2+^ concentration (Figure 4A) that resulted from Ca^2+^ influx and was not accompanied by IP_3_ formation, in contrast to the positive control ADH (Figure 4B–D). Preventing Ca^2+^ influx by lowering extracellular Ca^2+^ did not prevent aldosterone-induced cytosolic alkalinization (Figure 4E). Furthermore, buffering of intracellular Ca^2+^ did not prevent the aldosterone-induced increase in cell number (Figure 4F). 

## 4. Discussion

Our data show that aldosterone induces rapid cytosolic alkalinization in glomerular mesangial cells (Figure 5). This effect is independent of HCO_3_^−^, indicating that HCO_3_^−^ transports are not involved. The lack of effect of DIDS supports this conclusion. By contrast, the pharmacological data indicate that aldosterone stimulates Na^+^/H^+^ exchange in glomerular mesangial cells, similar to its effect in other cell systems [58]. Because (i) only isoform 1 of Na^+^/H^+^ exchange has been detected in rat glomerular mesangial cells [59] and (ii) EIPA at 10 µmol/L prevented alkalinization completely, we conclude that aldosterone stimulates Na^+^/H^+^-exchange type 1 (NHE1). Aldosterone-induced acidification could not be prevented by either EIPA or DIDS. At present, the mechanisms underlying acidification in GMC are not known, as is the case for other cell types, like vascular smooth muscle cells [21]. 

The inhibitory effect of U0126 on aldosterone-induced alkalinization, taken together with known aldosterone-induced ERK1/2 activation in GMC [10], indicates that the activation of NHE1 in rat mesangial cells by aldosterone is mediated by ERK1/2. Aldosterone-induced stimulation of NHE1 via ERK1/2 has also been reported for MDCK and M-1 cells [37,60]. Therefore, the effects observed in GMC and other cells seem to describe a mechanism of action of aldosterone with broader significance. Furthermore, we provide evidence for an involvement of protein kinase C (inhibitory effect of BIM) but not of IP3, indicating that the classical phospholipase Cβ is most probably not involved. Our data do make it possible to define the relevant PKC isoform nor to conclude whether PKC activity has to be increased or just exerts a permissive role. Although the effect of aldosterone is rapid, it seems to be mediated by the mineralocorticoid receptor, indicated by the inhibitory action of spironolactone, which is also responsible for a variety of non-genotropic aldosterone effects [31]. 

Because cell proliferation depends on a proper cytoplasmic pH homeostasis [32,33], the importance of aldosterone-induced alkalinization may arise from its contribution to cell growth. Our data on cell proliferation confirm the study by Nishiyama et al. [10], showing that aldosterone stimulates GMC proliferation via ERK1/2. Furthermore, our data show that this proliferation requires cytoplasmic alkalinization via NHE1 activation, because inhibition of NHE1 abrogates proliferation. Thus, our data add one additional piece to the puzzle of aldosterone-induced proliferation of GMC, which depends on ERK1/2 and NHE1. Finally, aldosterone-induced proliferation leads to enhanced collagen secretion. At present, we do not know whether a specific type of collagen is affected or whether it is an overall increase of collagen secretion, but we will address this issue in future studies. If the proliferation and collagen data also hold true for the in vivo situation, they would nicely help to explain sclerosis of glomeruli.

## 5. Conclusions

MR activation, e.g., by aldosterone, is part of progressive inflammatory renal damage independently of blood pressure [61], and the glomerular mesangium seems to be a major target. Our data suggest that the underlying pathomechanisms leading to glomerular mesangial injury include ERK1/2- and NHE1-induced proliferation of GMC and thereby to glomerular hypercellularity and dysfunction. Knowledge of the cellular mechanisms leads to a better understanding of the pathophysiology of renal injury and may help to develop therapeutic strategies.

## Figures and Tables

**Figure 1 biomedicines-09-01117-f001:**
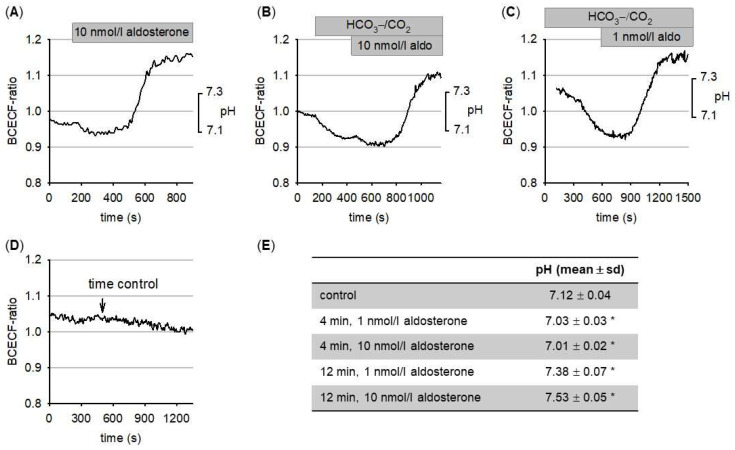
**(A**) Aldosterone induces a rapid and biphasic change in cytosolic pH. Transient acidification is followed by sustained alkalinization. (**B**) Aldosterone-induced pH-changes are also observed in HCO_3_^−^/CO_2_-buffered Ringer-solution. (**C**) pH-changes are also observed with 1 nmol/L aldosterone. (**D**) Time control experiments in HCO_3_^−^/CO_2_-buffered Ringer-solution show that cytosolic pH does not increase under control conditions. (**E**) Summary of the pH-values before addition of aldosterone (control), during maximum acidification (4 min) and when the plateau of alkalinization was reached (12 min). N = 70–120 for each condition. * = *p* < 0.05 versus control.

**Figure 2 biomedicines-09-01117-f002:**
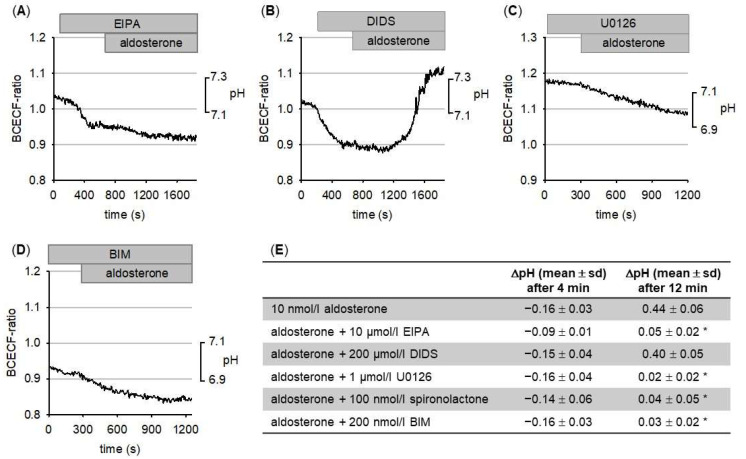
(**A**) The Na^+^/H^+^-exchange inhibitor EIPA prevents aldosterone-induced alkalinization. (**B**) Inhibition of HCO_3_^−^-transporters with DIDS does not prevent aldosterone-induced alkalinization. (**C**) Aldosterone-induced alkalinization is abolished when ERK1/2-activation is prevented by U0126. (**D**) Aldosterone-induced alkalinization is abolished by PKC inhibition with BIM. (**E**) Summary of the pH-changes under the different conditions. N = 60–100 for each condition. * = *p* < 0.05 versus control.

**Figure 3 biomedicines-09-01117-f003:**
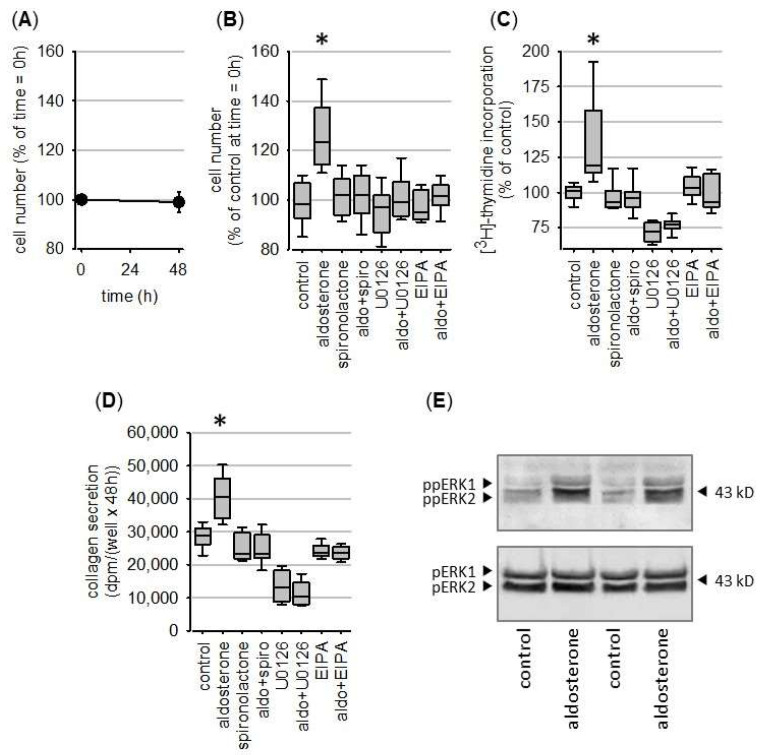
(**A**) Cell number is stable over a 48 h period in minus media. (**B**,**C**) Aldosterone (aldo, 10 nmol/L) induced proliferation of GMC proliferation is prevented by inhibition of Na^+^/H^+^-exchange (EIPA), by inhibition of ERK1/2-phosphorylation (U0126) and by blockade of the mineralocorticoid receptor (spiro). N = 10. * = *p* < 0.05 versus respective control. (**D**) Aldosterone (aldo, 10 nmol/L) induced proliferation collagen secretion is prevented by inhibition of Na^+^/H^+^-exchange (EIPA), by inhibition of ERK1/2-phosphorylation (U0126) and by blockade of the mineralocorticoid receptor (spiro). N = 8. * = *p* < 0.05 versus respective control. (**E**) Aldosterone (10 nmol/L, 15 min) enhances ERK1/2 phosphorylation (ppERK1/2).

**Figure 4 biomedicines-09-01117-f004:**
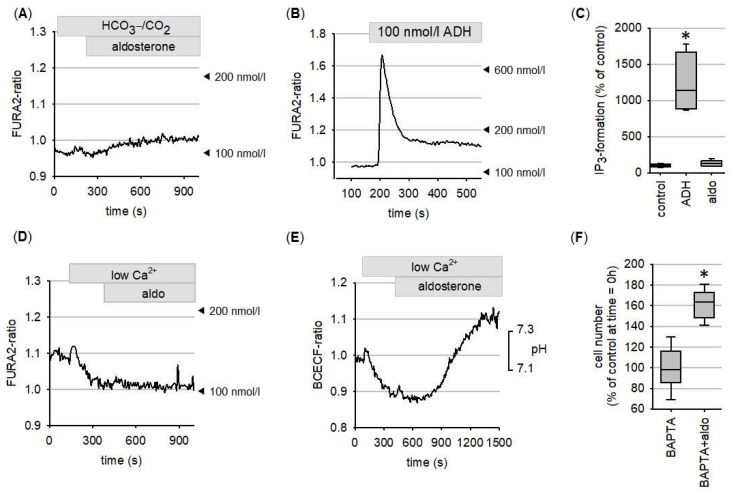
(**A**) Aldosterone (10 nmol/L) induces only a minor increase in cytosolic Ca^2+^, in contrast to (**B**) 100 nmol/L ADH. (**C**) ADH but not aldosterone induces the formation of IP_3_ (N = 5). (**D**) At low extracellular Ca^2+^, to prevent relevant Ca^2+^ influx, aldosterone induces no increase of cytosolic Ca^2+^. (**E**) Aldosterone-induced alkalinisation and (**F**) aldosterone-induced increase in cell number are independent of extracellular Ca^2+^ (N = 10). * = *p* < 0.05 versus respective control.

**Figure 5 biomedicines-09-01117-f005:**
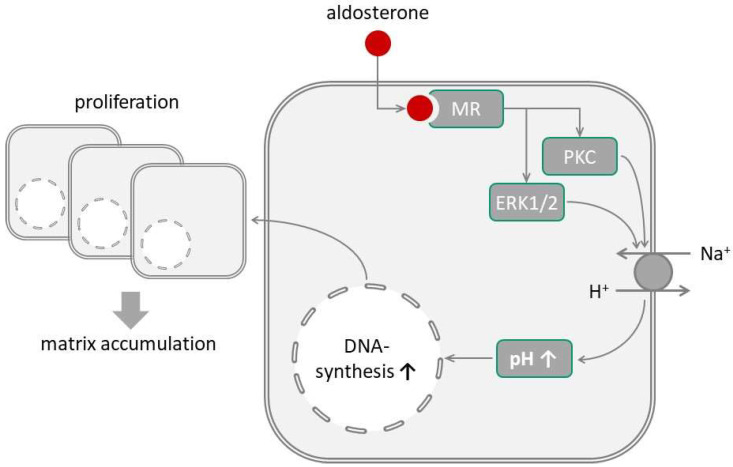
Graphical summary of the described effects of aldosterone. MR = mineralocorticoid receptor. PKC = protein kinase C. ERK1/2 = extracellular signal regulated kinase 1/2.

## Data Availability

Not applicable.

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
