# Peer review of "Glomerular Mesangial Cell pH Homeostasis Mediates Mineralocorticoid Receptor-Induced Cell Proliferation"

_biomedicines, 2021, doi:10.3390/biomedicines9091117_

Round 1

Reviewer 1 Report

In this study, authors used rat glomerular cells to clarify the rapid signaling pathway of aldosterone via MR-mediated activation of NHE and alkalinization, leading to cell proliferation. Mainly by using pharmacological inhibitors, they showed each pathway is involved in the mechanism of action.

  1. Figure 1D→Is this under HCO3-/CO2-buffered solution? It is better to show time control both with and without HCO3-/CO2-buffered solution.

  1. Please provide references to support that the concentration of each inhibitor is appropriate.

  1. As related above, it is better to show that MRA but not the other inhibitors block aldosterone mediated-transcriptional effect via MR as a nuclear receptor. This would help supporting the specificity of the inhibitors.

Author Response

1. Figure 1D→Is this under HCO3-/CO2-buffered solution? It is better to show time control both with and without HCO3-/CO2-buffered solution.

The time control was performed in HCO3-/CO2-buffered Ringer-solution, which represents the physiological situation more closely than HEPES-buffered Ringer solution. This is now mentioned in the figure legend.

2. Please provide references to support that the concentration of each inhibitor is appropriate.

The references are now included in the manuscript.

EIPA: [1,2]. U0126: [3]. DIDS: [4]. BIM: [5,6]

3. As related above, it is better to show that MRA but not the other inhibitors block aldosterone mediated-transcriptional effect via MR as a nuclear receptor. This would help supporting the specificity of the inhibitors.

We reported previously that BIM (Pfau et al., MCE 2006, [7]) exerts no effect on MR transcriptional activity. By contrast, ERK1/2 is required for regular nuclear translocation (Grossmann et al., ME 2005, [8]), which leads to the modulation of transcriptional activity.  By contrast, DIDS has been shown not to prevent transcriptional effects of aldosterone (AJP, 2006 Mar; 290(3):C757-6, [9]). Similarly, EIPA did not prevent MR-induced upregulation of sodium channel expression (Steroids, 1995; 60(1):114-9, [10]).

These informations are now included in the manuscript.

Reference List

  1. Noel, J. and Pouysségur, J. Hormonal regulation, pharmacology, and membrane sorting of vertebrate Na+/H+exchanger isoforms. Am.J.Physiol. 1995, 268, C283-C296.
  2. Masereel, B.; Pochet, L. and Laeckmann, D. An overview of inhibitors of Na+/H+ exchanger. European Journal of Medicinal Chemistry 2003, 38, 547-554.
  3. Favata, M.F.; Horiuchi, K.Y.; Manos, E.J.; Daulerio, A.J.; Stradley, D.A.; Feeser, W.S.; Van Dyk, D.E.; Pitts, W.J.; Earl, R.A.; Hobbs, F.; Copeland, R.A.; Magolda, R.L.; Scherle, P.A. and Trzaskos, J.M. Identification of a Novel Inhibitor of Mitogen-activated Protein Kinase Kinase *. J.Biol.Chem. 1998, 273, 18623-18632.
  4. Emmons, C. Transport characteristics of the apical anion exchanger of rabbit cortical collecting duct ß-cells. American Journal of Physiology-Renal Physiology 1999, 276, F635-F643.
  5. Martiny-Baron, G.; Kazanietz, M.G.; Mischak, H.; Blumberg, P.M.; Kochs, G.; Hug, H.; Marmé, D. and Schächtele, C. Selective inhibition of protein kinase C isozymes by the indolocarbazole Gö 6976. J Biol Chem. 1993, 268, 9194-9197.
  6. Gschwendt, M.; Dieterich, S.; Rennecke, J.; Kittstein, W.; Mueller, H.J. and Johannes, F.J. Inhibition of protein kinase C by various inhibitors. Inhibition from protein kinase c isoenzymes. FEBS Letters 1996, 392, 77-80.
  7. Pfau, A.; Grossmann, C.; Freudinger, R.; Mildenberger, S.; Benesic, A. and Gekle, M. Ca2+ but not H2O2 modulates GRE-element activation by the human mineralocorticoid receptor in HEK cells. Mol.Cell.Endocrinol. 2007, 264, 35-43.
  8. Grossmann, C.; Benesic, A.; Krug, A.W.; Freudinger, R.; Mildenberger, S.; Gassner, B. and Gekle, M. Human mineralocorticoid receptor expression renders cells responsive for nongenotropic aldosterone actions. Molecular Endocrinology 2005, 19, 1697-1710.
  9. Good, D.W.; George, T. and Watts, B.A. Nongenomic regulation by aldosterone of the epithelial NHE3 Na+/H+ exchanger. American Journal of Physiology-Cell Physiology 2006, 290, C757-C763.
  10. Kornel, L. and Smoszna-Konaszewska, B. Aldosterone (ALDO) increases transmembrane influx of Na+ in vascular smooth muscle (VSM) cells through increased synthesis of Na+ channels. Steroids 1995, 60, 114-119.

Reviewer 2 Report

In this study, the authors investigated the glomerular mesangial cell pH-homeostasis mediates mineralo-corticoid receptor-induced cell proliferation. The results showed aldosterone can induce glomerular mesangial cell (GMC) proliferation via a MR and ERK1/2-mediated activation of NHE with subsequent cytosolic alkalinization. This study is very interesting and I only have two minor concerns listed below.

  • About the concentration of aldosterone, some experiment used 1 nmol/l and others used 10 nmol/l. How to determine the concentration for the specific experiment?
  • In figure 3E, the total expression of ERK1/2 should be shown as a control. And the molecular weight also should be shown in immunoblot.

Author Response

About the concentration of aldosterone, some experiment used 1 nmol/l and others used 10 nmol/l. How to determine the concentration for the specific experiment?

Initially, we performed experiments in parallel with 1 and 10 nmol/l aldosterone (Fig. 1) and observed similar effects, with a slightly larger effect size for 10 nmol/l. For the subsequent working packages we used 10 nmol/l aldosterone.

In figure 3E, the total expression of ERK1/2 should be shown as a control. And the molecular weight also should be shown in immunoblot.

We apologize fort he mislabeling. The figure was corrected and now shows phosphorylated and total pERK1/2 as well as a molecular weight indicator.

Round 2

Reviewer 1 Report

I am satisfied with the authors' responses.